# Limbic System Response to Psilocybin and Ketamine Administration in Rats: A Neurochemical and Behavioral Study

**DOI:** 10.3390/ijms25010100

**Published:** 2023-12-20

**Authors:** Adam Wojtas, Agnieszka Bysiek, Agnieszka Wawrzczak-Bargiela, Marzena Maćkowiak, Krystyna Gołembiowska

**Affiliations:** 1Unit II, Department of Pharmacology, Maj Institute of Pharmacology, Polish Academy of Sciences, 12 Smętna Street, 31-343 Kraków, Poland; wojtas@if-pan.krakow.pl (A.W.); bysiek@if-pan.krakow.pl (A.B.); 2Laboratory of Pharmacology and Brain Biostructure, Department of Pharmacology, Maj Institute of Pharmacology, Polish Academy of Sciences, 12 Smętna Street, 31-343 Kraków, Poland; bargiela@if-pan.krakow.pl (A.W.-B.); mackow@if-pan.krakow.pl (M.M.)

**Keywords:** neurotransmitter release, dopamine D2 receptors, serotonin 5-HT1A, 5-HT2A receptors, limbic system

## Abstract

The pathophysiology of depression is related to the reduced volume of the hippocampus and amygdala and hypertrophy of the nucleus accumbens. The mechanism of these changes is not well understood; however, clinical studies have shown that the administration of the fast-acting antidepressant ketamine reversed the decrease in hippocampus and amygdala volume in depressed patients, and the magnitude of this effect correlated with the reduction in depressive symptoms. In the present study, we attempted to find out whether the psychedelic substance psilocybin affects neurotransmission in the limbic system in comparison to ketamine. Psilocybin and ketamine increased the release of dopamine (DA) and serotonin (5-HT) in the nucleus accumbens of naive rats as demonstrated using microdialysis. Both drugs influenced glutamate and GABA release in the nucleus accumbens, hippocampus and amygdala and increased ACh levels in the hippocampus. The changes in D2, 5-HT1A and 5-HT2A receptor density in the nucleus accumbens and hippocampus were observed as a long-lasting effect. A marked anxiolytic effect of psilocybin in the acute phase and 24 h post-treatment was shown in the open field test. These data provide the neurobiological background for psilocybin’s effect on stress, anxiety and structural changes in the limbic system and translate into the antidepressant effect of psilocybin in depressed patients.

## 1. Introduction

As reported by the World Health Organization, major depressive disorder can affect up to 5% of the adult population worldwide [1]. In comparison, 10 to 15% of adults experience a depressive episode during their lifetime, although these statistics can be underestimated because many individuals suffering from affective disorders do not seek professional help [2]. Even though the pathophysiology of depression is still not clearly understood, several key factors relevant to this problem have been proposed. Firstly, numerous risk factors have been mentioned, with the most significant being either traumatic or chronic stress [3]. Moreover, depression-related changes have been observed in various brain structures, most notably in the frontal cortex and the limbic system [4]. However, the direction of changes was opposite: namely, the hypoactivity was shown in the cortex while hyperactivity was shown in the limbic structures [5]. The prefrontal cortex appears to be a center that integrates many sensory inputs from structures like the hippocampus, amygdala and nucleus accumbens, assigning them either rewarding or aversive properties [3].

The most Important component of the limbic system that is affected in major depressive disorder is the hippocampus [4]. Like in the case of the prefrontal cortex, studies showed a clear correlation between atrophy of the hippocampus and the severity and length of the disorder [3]. Patients with greater hippocampal volume seem to respond better to the treatment, while those with a smaller volume are more prone to treatment resistance [4]. Those structural changes translate into functional ones, as the hippocampus plays a crucial role in the processes of learning and memory [5] and is one of a few places where neurogenesis can occur in adults.

The amygdala plays a central role in processes associated with emotional states and fear processing. As with the hippocampus, its reduced volume correlates with the severity of the depression [6]; moreover, its hyperactivity correlates with the severity of the disorder [5].

In contrast to the two aforementioned brain areas, the nucleus accumbens exhibit hypertrophy both in preclinical models and in individuals suffering from depression, which probably arises from the observed increase in the length of the dendrites and spine density [7]. Furthermore, according to DSM-V [8], anhedonia is one of the two core symptoms of depression, while the inability to experience pleasure is a clear manifestation of disorders in the functioning of the reward system, where the nucleus accumbens play a crucial role [9].

Recent studies showed that ketamine, an N-methyl-D-aspartate (NMDA) receptor antagonist proposed as a prototypical fast-acting antidepressant drug [10,11], exerts its beneficial effect not only in the frontal cortex but also by targeting areas in the limbic system. The administration of ketamine promotes neural plasticity in the hippocampus in preclinical models [12] and increases hippocampal volume in depressed patients [13] while significantly improving their mood [13,14], which is an effect correlated with the volume of the hippocampus. Zhou et al. [15] observed that ketamine administration increased the amygdala volume in depressed patients, and the magnitude of the effect correlated with the reduction in depressive symptoms. Furthermore, ketamine treatment seems to reverse the depression-induced hypertrophy of the nucleus accumbens [7]. Altogether, the research cited above leads to the conclusion that ketamine affects not only the prefrontal cortex but also the limbic structures, inducing a complex effect that normalizes its functioning and increases its connectivity with the prefrontal cortex [3].

Despite opening a new chapter in the treatment of affective disorders and becoming the prototypical fast-acting antidepressant drug, ketamine is far from perfect. Its effects fade around the second week after the drug administration, leading to the need for repeated dosing [16]. Furthermore, it possesses a plethora of adverse effects, of which drug abuse is the most significant one. Similarly to ketamine, psychedelics seem to induce synaptic plasticity and neurogenesis in preclinical models [16,17] while being devoid of the drawbacks of ketamine. This phenomenon translates further into antidepressant effects observed in preclinical models after the administration of psychedelics [18,19]. Among serotonergic hallucinogens, psilocybin is currently the most studied compound, as it has proven to be at least as effective as common antidepressant drugs [20]. In contrast to ketamine, reports on the possible effects exerted by psychedelics on the limbic system are scarce, justifying the need for thorough studies.

To address the questions regarding the response of the limbic system to the administration of psilocybin and to compare it to ketamine, we have examined the effects of both drugs on neurotransmission in the hippocampus, amygdala and nucleus accumbens using microdialysis in freely moving rats. Furthermore, Western blot analysis was performed to assess the long-lasting effects of the chosen drugs on protein levels of selected receptors. As the limbic system plays a crucial role in locomotion and fear response, the rat behavior was studied in the open field test both directly and 24 h after administration of the drugs to examine both acute and possible prolonged effects.

## 2. Results

### 2.1. The Effect of Psilocybin and Ketamine on Extracellular Levels of DA and 5-HT in the Rat Nucleus Accumbens

Psilocybin at doses of 2 and 10 mg/kg significantly increased extracellular levels of DA up to ca. 180% of baseline in the rat nucleus accumbens (Figure 1A). Ketamine (10 mg/kg) was more potent in increasing (up to ca. 250% of baseline) DA extracellular level (Figure 1A). Repeated measures ANOVA showed a significant effect of treatment groups (F_3,23_ = 123, *p* < 0.0001), sampling period (F_11,253_ = 15.2, *p* < 0.0001), and the interaction between treatment groups and sampling period (F_33,253_ = 15.9, *p* < 0.0001). Total effects expressed as AUC shown in Figure 1B were significantly increased for psilocybin 2 and 10 mg/kg and ketamine; furthermore, the effect of the ketamine group was significantly stronger than that of the psilocybin groups (F_3,23_ = 124, *p* < 0.001, one-way ANOVA).

The extracellular 5-HT level was increased in the rat nucleus accumbens by both doses of psilocybin (up to 200–250% of baseline) and less potently by ketamine (up to 200% of baseline) (Figure 1C). Repeated-measures ANOVA showed a significant effect of treatment groups (F_3,20_ = 267, *p* < 0.0001), sampling period (F_11,220_ = 23, *p* < 0.0001), and the interaction between treatment groups and sampling period (F_33,220_ = 16.1, *p* < 0.0001). The total effects expressed as AUC shown in Figure 1D were significantly increased for both psilocybin doses, 2 and 10 mg/kg, and for ketamine; the effect of the ketamine group was significantly different only in comparison to psilocybin at the dose of 10 mg/kg (F_3,20_ = 268, *p* < 0.0001, one-way ANOVA).

### 2.2. The Effect of Psilocybin and Ketamine on Extracellular Levels of Glutamate and GABA in the Rat Nucleus Accumbens, Hippocampus and Amygdala

The extracellular glutamate (GLU) level in the nucleus accumbens was slightly but significantly decreased by both psilocybin doses (to ca. 80% of baseline), but it was markedly increased by ketamine (up to 160% of baseline) (Figure 2A). Repeated measures ANOVA showed a significant effect of treatment groups (F_3,26_ = 99, *p* < 0.0001), sampling period (F_11,286_ = 32, *p* < 0.0001), and the interaction between treatment groups and sampling period (F_33,286_ = 8.7, *p* < 0.0001). The total effects expressed as AUC shown in Figure 2B were significantly decreased by psilocybin 2 mg/kg and 10 mg/kg and increased by ketamine; the ketamine group was significantly different from both psilocybin groups (F_3,26_ = 99, *p* < 0.0001, one-way ANOVA).

The extracellular level of GABA in the nucleus accumbens was slightly and not significantly increased by a psilocybin dose of 2 mg/kg (up to ca. 120% of baseline), while it was more potently affected by the higher dose of 10 mg/kg (up to ca. 150% of baseline) and by ketamine (up to 180% of baseline) (Figure 3A). Repeated measures ANOVA showed a significant effect of treatment groups (F_3,25_ = 62, *p* < 0.0001), sampling period (F_11,275_ = 12.8, *p* < 0.0001), and the interaction between treatment groups and sampling period (F_33,275_ = 7.9, *p* < 0.0001). The total effects expressed as AUC shown in Figure 3B were significantly increased by the higher dose of psilocybin and ketamine; the ketamine effect was significantly different from the psilocybin at the dose of 2 mg/kg (F_3,35_ = 276, *p* < 0.0001, one-way ANOVA).

The extracellular glutamate (GLU) level in the hippocampus was significantly decreased by a psilocybin dose of 2 mg/kg (up to ca. 50% of baseline) but was markedly increased by the higher dose of 10 mg/kg (up to ca. 150% of baseline) and by ketamine (up to 140% of baseline) (Figure 2C). Repeated measures ANOVA showed a significant effect of treatment groups (F_3,22_ = 365, *p* < 0.0001), sampling period (F_11,242_ = 19.4, *p* < 0.0001), and the interaction between treatment groups and sampling period (F_33,242_ = 8.7, *p* < 0.0001). The total effects expressed as AUC shown in Figure 2D were significantly decreased for psilocybin 2 mg/kg and significantly increased for psilocybin 10 mg/kg and ketamine; the ketamine group was significantly different from psilocybin in the 2 mg/kg group (F_3,22_ = 366, *p* < 0.0001, one-way ANOVA).

The extracellular level of GABA in the hippocampus was significantly increased by a psilocybin dose of 2 mg/kg (up to ca. 120% of baseline), while it was more potently affected by the higher dose of 10 mg/kg (up to ca. 140% of baseline) and by ketamine (up to 140% of baseline) (Figure 3C). Repeated measures ANOVA showed a significant effect of treatment groups (F_3,22_ = 223, *p* < 0.0001), sampling period (F_11,242_ = 16.8, *p* < 0.0001), and the interaction between treatment groups and sampling period (F_33,242_ = 8.5, *p* < 0.0001). Both psilocybin doses and ketamine significantly increased total effects expressed as AUC, as shown in Figure 3D; the ketamine effect was significantly different from the effect of the lower dose of psilocybin (F_3,22_ = 224, *p* < 0.0001, one-way ANOVA).

The extracellular glutamate (GLU) level in the amygdala was significantly increased by a psilocybin dose of 2 mg/kg (up to ca. 140% of baseline) and by ketamine (up to ca. 160% of baseline) between 20 and 100 min of the collection period and was not affected by both psilocybin doses and ketamine for the rest of the time (Figure 2G,E). Repeated measures ANOVA showed a significant effect of treatment groups (F_3,18_ = 20.5, *p* < 0.0001), sampling period (F_4,12_ = 15.4, *p* < 0.0001), and the interaction between treatment groups and sampling period (F_4,72_ = 11.6, *p* < 0.0001). The total effects expressed as AUC between 20 and 100 min of the collection period and shown in Figure 2H were significantly increased for psilocybin 2 mg/kg and ketamine and not changed for psilocybin 10 mg/kg; the ketamine effect was significantly different from the effect of a higher dose of psilocybin (F_3,18_ = 20.5, *p* < 0.0001, one-way ANOVA). The total effects expressed as AUC for the whole collection period did not differ from the control group (Figure 2F).

The extracellular level of GABA in the amygdala was slightly but not significantly increased by a psilocybin dose of 2 mg/kg and ketamine, and it was more potently affected by a higher psilocybin dose of 10 mg/kg (up to ca. 120% of baseline) (Figure 3G) between 20 and 100 min of the collection period, while it was not affected by both psilocybin doses and ketamine for the rest of the time (Figure 3E). Repeated measures ANOVA showed a significant effect of treatment groups (F_3,18_ = 3.2, *p* < 0.05), sampling period (F_4,12_ = 2.9, *p* < 0.03), and the interaction between treatment groups and sampling period (F_4,72_ = 4.9, *p* < 0.001). Total effects expressed as an AUC between 20 and 100 min of the collection period and shown in Figure 3H were significantly increased for the psilocybin dose of 10 mg/kg and were not changed for the psilocybin dose of 2 mg/kg and ketamine; there were no differences observed between ketamine and psilocybin doses (F_3,18_ = 3.2, *p* < 0.05, one-way ANOVA). Total effects expressed as AUC for the whole collection period did not differ from the control group despite ketamine; the ketamine group effect was significantly increased in comparison to psilocybin at the higher dose (F_3,18_ = 6.5, *p* < 0.004, one-way ANOVA) (Figure 3F).

### 2.3. The Effect of Psilocybin and Ketamine on GABA/Glutamate Ratio in the Rat Nucleus Accumbens, Hippocampus and Amygdala

To find out the net effect of psilocybin and ketamine on GABA and glutamate release in rat brain regions, the GABA/GLU ratio was calculated. The mean of the GABA/GLU index of AUC values for each group is presented in Figure 4A. There was an increase in the GABA/GLU ratio for the whole collection period for psilocybin doses of 2 and 10 mg/kg but not for ketamine in the nucleus accumbens (F_3,25_ = 31, *p* < 0.0001, one-way ANOVA). The increase in GABA/GLU ratio of AUC values for the 2 mg/kg psilocybin group but not for the higher dose and ketamine was observed in the hippocampus; the ketamine group effect was significantly different in comparison to the 10 mg/kg psilocybin group (Figure 4A) (F_3,25_ = 255, *p* < 0.0001, one-way ANOVA). No change was found in the GABA/GLU ratio of AUC values for the whole collection period in the amygdala (Figure 4A) (F_3,18_ = 0.96, *p* < 0.43, one-way ANOVA). However, the decrease in the GABA/GLU ratio of AUC values for the 2 mg/kg psilocybin group and ketamine was observed between 20 and 100 min of the collection period; the ketamine group effect was significantly different from the effect of a higher dose of psilocybin (Figure 4B) (F_3,18_ = 8.3, *p* < 0.001, one-way ANOVA).

### 2.4. The Effect of Psilocybin and Ketamine on Extracellular Levels of ACh in the Rat Hippocampus

The extracellular ACh level was increased in the rat hippocampus by both doses of psilocybin and ketamine (up to 700%, 200% and 250% of baseline, respectively) (Figure 5A). Repeated measures ANOVA showed a significant effect of treatment groups (F_3,18_ = 74, *p* < 0.0001), sampling period (F_5,90_ = 92, *p* < 0.0001), and the interaction between treatment groups and sampling period (F_15,90_ = 35, *p* < 0.0001). Total effects expressed as AUC shown in Figure 5B were significantly increased for both psilocybin doses, 2 and 10 mg/kg, and for ketamine; the ketamine effect was significantly different from that of the 2 mg/kg psilocybin group (F_3,18_ = 74, *p* < 0.0001, one-way ANOVA).

### 2.5. The Effect of Psilocybin and Ketamine on 5-HT1A and 5-HT2A Receptor Levels in the Rat Hippocampus

Both doses of psilocybin significantly decreased the 5-HT1A protein level by ca. 12–13%, while ketamine significantly increased it by 18% over control; the ketamine effect was significantly different from both psilocybin groups (F_3,28_ = 60, *p* < 0.0001, one-way ANOVA) as measured 7 days after drug administration (Figure 6A,B). The 5-HT2A protein level was decreased by the psilocybin 2 mg/kg dose, but it was increased by the higher one and ketamine by 11 and 35%, respectively, in comparison to control; the ketamine effect was significantly higher than that of both psilocybin groups (F_3,28_ = 57, *p* < 0.001, one-way ANOVA) (Figure 6C,D).

### 2.6. The Effect of Psilocybin and Ketamine on D2 and 5-HT2A Receptor Levels in the Rat Nucleus Accumbens

In the nucleus accumbens, the dopamine D2 protein receptor level was significantly increased by the higher dose of psilocybin by ca. 30% but not by its lower dose or ketamine, while the ketamine effect was significantly different from that of the 10 mg/kg psilocybin group (F_3,28_ = 11.6, *p* < 0.0001, one-way ANOVA) as measured 7 days after drug administration (Figure 7C,D). The 5-HT2A protein level was not affected by any treatment (F_3,28_ = 0.24, *p* < 0.87, one-way ANOVA) (Figure 7A,B).

### 2.7. The Effect of Psilocybin and Ketamine on the Activity of Rats in the Open Field Test

Psilocybin at doses of 2 and 10 mg/kg significantly decreased the time of walking. In contrast, ketamine (10 mg/kg) increased it above the control level, and its effect was significantly different from both psilocybin groups one hour after drugs administration (Figure 8A) (F_3,36_ = 47, *p* < 0.001, one-way ANOVA). Both doses of psilocybin significantly decreased the number of crossings reflecting ambulatory distance, which was not changed by ketamine, but its effect was significantly different from the psilocybin groups (Figure 8A) (F_3,36_ = 44, *p* < 0.0001, one-way ANOVA). The number of episodes of peeping and rearing reflecting vertical activity was decreased by both doses of psilocybin and not changed by ketamine one hour after administration; however, there was a significant difference between ketamine and psilocybin groups in the number of peeping episodes (Figure 8B) (F_3,36_ = 10.1, *p* < 0.001; F_3,36_ = 4.7, *p* < 0.01, one-way ANOVA, respectively). The center exploration was significantly increased by both doses of psilocybin but not by ketamine in comparison to the control. A significant difference between the ketamine and psilocybin groups was observed one hour after drug administration (Figure 8B) (F_3,36_ = 98, *p* < 0.0001, one-way ANOVA).

The time of walking and the number of crossings reflecting ambulatory distance were not changed by both doses of psilocybin and ketamine in comparison to the control 24 h after drug administration (Figure 8C) (F_3,36_ = 0.49, *p* < 0.69; F_3,36_ = 0.02, *p* < 0.99, one-way ANOVA, respectively). Similarly, the number of episodes of rearing was not changed by both doses of psilocybin and ketamine 24 h after drug administration (Figure 8D) (F_3,36_ = 1.87, *p* < 0.15, one-way ANOVA). However, the number of episodes of peeping was decreased by the psilocybin dose of 2 mg/kg and ketamine 24 h after drug administration; there was a significant difference observed between ketamine and a higher dose of psilocybin (Figure 8D) (F_3,36_ = 12.2, *p* < 0.0001, one-way ANOVA). The center exploration was significantly increased by both doses of psilocybin and ketamine, and its effect was significantly higher in comparison to psilocybin doses 24 h after drug administration (Figure 8D) (F_3,36_ = 29, *p* < 0.0001, one-way ANOVA).

Animals that underwent the EPM test (Appendix A) did not display anxiety-like behavior when measured 1 h after preexposure to an open field test.

## 3. Discussion

The data from numerous empirical studies support the idea that fast-acting psychedelics enable signaling within anatomical networks essential for a range of cognitive and affective tasks. The antidepressant properties of psilocybin are mediated via modulation of the prefrontal and limbic regions [21]. The molecular target of psilocybin in the brain was identified as the serotonin 5-HT2A receptor expressed in the subset of deep pyramidal cells in layer V of the prefrontal cortex [22]. The interaction of psilocybin with 5-HT2A receptors produces psychomimetic effects and a rise in glutamate levels, as confirmed in clinical studies and animal models [23,24]. The acute activation of glutamate neurotransmission was associated with an upregulation of BDNF and subsequent synaptic plasticity [3,21]. Intriguingly, depression and chronic stress increase cortical and hippocampal extracellular glutamate and excitotoxicity, subsequently precipitating neuronal atrophy [25]. Thus, the question arises regarding whether the antidepressant effects of rapid-acting psychedelics are due to the inhibition of glutamate neurotransmission. Psilocybin also interacts with 5-HT1A receptors although with lower affinity [26,27]. Excitatory 5-HT2A and inhibitory 5-HT1A receptors colocalize in both cortical pyramidal neurons and GABAergic interneurons; thus, the cellular response is determined by the summation of 5-HT1A inhibition and 5-HT2A excitation [28]. Through the modulation of cellular excitability, psilocybin may impact cortical projections to other brain circuits such as the nucleus accumbens, hippocampus and amygdala [29]. Thus, glutamatergic and GABAergic signaling would be implicated in cortico-limbic function.

Psilocybin exhibits no affinity for dopamine D2 receptors [21], but it interacts indirectly with mesolimbic dopaminergic pathways, which play a significant role in the brain reward system [30]. A low addictive potential of psilocybin suggests this proposed indirect mechanism of action. Furthermore, there is a positive correlation between depression and dopamine deficiency [31].

In our work, we demonstrated a dose-dependent increase in extracellular levels of DA elicited by psilocybin. The regulation of mesocortical DA by psilocybin may involve cortical glutamatergic fibers expressing 5-HT2A receptors projecting to the nucleus accumbens or ventral tegmental area (VTA), thus indirectly increasing DA release. On the other hand, also 5-HT1A receptors seem to indirectly alter DA release due to a low density of 5-HT1A receptors in the nucleus accumbens. However, 5-HT1A receptors may control DA release by reducing the 5-HT neuron activity as a consequence of the stimulation of 5-HT1A autoreceptors in the dorsal raphe nucleus or by reducing pyramidal cell activity projecting to VTA neurons [32]. A similar effect on DA release in the nucleus accumbens exerted by psilocin, an active metabolite of psilocybin, was demonstrated by Sakashita et al. [33].

Ketamine, a fast-acting antidepressant used in our study for comparison, also elevated DA release with a potency similar to the higher dose of psilocybin. However, the mechanism of ketamine action in the regulation DA release in the nucleus accumbens differs from that of psilocybin, since it is mediated through the disinhibition of glutamatergic fibers projecting to the VTA by blocking NMDA receptors located in cortical GABAergic interneurons [34].

The excitatory effect of psilocybin on cortical pyramidal neurons projecting to the dorsal raphe may be responsible for a dose-dependent increase in 5-HT release in the nucleus accumbens. However, 5-HT release from serotonin terminals may also be modulated by 5-HT2A receptors located in GABAergic interneurons in the dorsal raphe cells, but this mechanism seems to be less pronounced since the levels of 5-HT2A receptor mRNA in the dorsal raphe cells are low [35]. The weaker effect of ketamine on serotonin release as compared to psilocybin may involve NMDA receptors in GABAergic neurons disinhibiting glutamatergic innervation of dorsal raphe cells [28]. In addition, AMPA receptors might be also involved in the behavioral and neurochemical effects of ketamine [36]. What is more, ketamine might enhance serotonergic transmission by the inhibition of SERT activity [37].

Intriguingly, preclinical studies demonstrated hypertrophy of the nucleus accumbens in models of depression [25]. It is suggested that the stress-induced nucleus accumbens hypertrophy may be related to dopaminergic neurotransmission abnormalities in the VTA pathway to the nucleus accumbens [38]. Furthermore, stress and depression are believed to precipitate phasic activation of the VTA–nucleus accumbens pathway, leading to DA and BDNF release in the nucleus accumbens [39]. Subsequently, the stress-induced BDNF release results in nucleus accumbens hypertrophy and depressive-like behavior [40]. Our experiments showing the stimulation of DA release in the nucleus accumbens by psilocybin do not seem to explain the above-mentioned observations. However, the effect of psilocybin in naive rats may differ from its action in animals exposed to stress. Further studies are necessary to find out whether fast-acting antidepressants are able to normalize stress-induced abnormalities in the nucleus accumbens observed in rodents and depressive patients [7].

Our data show another pattern of psilocybin action on extracellular glutamate levels in limbic regions. The direct stimulation of 5-HT1A receptors located on the pyramidal cells or 5-HT2A receptors on GABAergic interneurons by psilocybin, reducing the prefrontal cortex output to the nucleus accumbens [28,29], may be responsible for the decrease in glutamate release in the nucleus accumbens. A lack of dose–response linearity in this effect stems from differences in the density of receptor subtypes in both locations [26,28]. Ketamine used in our study as a comparator significantly increased glutamate levels in the nucleus accumbens. The ketamine-induced enhancement of glutamate release is likely mediated via its blocking activity on NMDA receptors within GABAergic interneurons, resulting in the disinhibition of pyramidal cells projecting to the nucleus accumbens [41].

Moreover, psilocybin and ketamine increased GABA release in this region. GABAergic neurons are widely distributed in the shell and core of the nucleus accumbens. They bear 5-HT1A and 5-HT2A receptor subtypes, although they differ in their density [42]. The direct activation of inhibitory or excitatory subtypes may depend on the psilocybin dose. In addition, cortical projections to the nucleus accumbens may also influence the excitability of GABAergic neurons regulating GABA levels via other receptor types [43]. The effect on cortical projection by NMDA receptor blockade may explain a possible mechanism of ketamine action on GABA release in the nucleus accumbens [41].

A diverse psilocybin effect on glutamate release was found in the hippocampus. Its lower dose decreased while the higher one increased glutamate levels. In contrast, GABA release was dose-dependently enhanced by psilocybin. The observed changes in glutamate and GABA release depend on the stimulation of hippocampal 5-HT1A and 5-HT2A receptors by psilocybin. Both receptor subtypes have various distribution and density patterns in the hippocampus. 5-HT1A receptors are highly expressed on both principal glutamatergic cells and GABAergic interneurons [44]. The activation of 5-HT1A receptors primarily leads to the inhibition of hippocampal pyramidal cells [45]. This may be the cause of the decrease in glutamate release by the lower dose of psilocybin in our study. However, the activation of 5-HT1A receptors expressed on GABAergic interneurons would disinhibit principal glutamatergic cells and thus would counteract the direct effect of 5-HT1A receptors expressed on principal neurons. Additionally, 5-HT2A receptors are expressed on both principal glutamatergic cells and on different subtypes of hippocampal interneurons although in a lower density than 5-HT1A receptors [46]. Thus, since 5-HT2A receptors are stimulatory and are expressed on both principal cells and GABAergic interneurons, it could be expected that they would have mixed effects on the both cell types. This mechanism may underlie the stimulatory effect of the higher psilocybin dose on glutamate release and both doses on GABA release in the hippocampus. However, it has to be noted that the resultant effect of the lower psilocybin dose on amino acid neurotransmission is inhibitory. The activation of corticolimbic excitatory projections, either through the selective antagonism of inhibitory interneurons or cortical disinhibition by ketamine, is a probable mechanism of the increase in glutamate and GABA extracellular levels in the hippocampus [36]. Considering the reduced hippocampal volume in depressed patients resulting from hippocampal stress-induced neuronal atrophy and low levels of GABA and glutamate demonstrated in depressed patients [7,25,47], the normalization of amino acid neurotransmission and hippocampal volume by psilocybin may be expected.

The additional beneficial effects of psilocybin in depressed patients with cognitive impairments may be related to its stimulatory effect on Ach levels as found in our study. Ach appears to act as a neuromodulator in the brain, and its role is to change neuronal excitability, alter the presynaptic release of neurotransmitters and coordinate the firing of groups of neurons. ACh contributes also to synaptic plasticity [48]. The primary source of cholinergic innervation of the hippocampus derives from the basal forebrain cholinergic system [49]. Generally, 5-HT exerts a stimulatory influence on the release of ACh; however, the effect depends on mediation via particular serotonin receptor subtypes. The 5-HT1A receptor subtype mediates a stimulatory effect on ACh release in the hippocampus as shown in the hippocampal perfusate of conscious freely moving rats [50]. However, the systemic administration of the 5-HT2A receptor agonist DOI in a high dose also increased ACh release in the hippocampus, but mescaline, a potent 5-HT2A agonist, did not affect ACh release [51]. These findings are in line with our data and support the idea that both receptor subtypes seem to be involved in the stimulatory effect of psilocybin on ACh release in the hippocampus. However, the lack of linearity in the dose–response effect needs further studies. The recent data of Pacheco et al. [52] showing an enhancement of cognitive flexibility in rats by psilocybin may correlate with the psilocybin effect on ACh release in the hippocampus. The role of NMDA receptors in regulating ACh release cannot be excluded, as ketamine also increased Ach release with a strength similar to the psilocybin higher dose.

Abnormally high amygdala reactivity to negative affective stimuli has been implicated in the pathophysiology of depression [53]. Negative affect and amygdala response were reduced by psilocybin, suggesting that psilocybin may increase emotional and brain plasticity [54]. In our study, psilocybin and ketamine effects on extracellular glutamate and GABA levels were of short duration, and the overall effect shown as a GABA/glutamate index was stimulatory. Because a prolonged anxiety induced by chronic stress in mice causes the dysfunction of basolateral amygdala projection neurons [55] and that reduced amygdala volume correlated with the severity of depression [6], our data indicate that the facilitation of synaptic transmission from the prefrontal cortex to the amygdala and especially the predominant role of glutamatergic pathways may be the underlying mechanism of the beneficial effect of psilocybin and ketamine in treatment of anxiety and depressive states.

The low basal levels of DA and 5-HT in dialysates from hippocampal and amygdalar regions did not allow for the measurement of these monoamine levels after the administration of psilocybin and ketamine (see Deurwardère and Giovanni [32]). It has to be noted that the problems with the detection of hippocampal and amygdalar DA and 5-HT were potentially due to the shorter lengths of microdialysis probes capturing less monoamines from the surrounding tissue.

In order to assess the long-lasting effects of psilocybin, we examined the density level of several receptor subtypes in the limbic system seven days after the cessation of treatments. The increase in dopamine D2 receptor density in the nucleus accumbens by the higher psilocybin dose may be a subsequent effect to the increased DA release in the nucleus accumbens and the regulatory mechanism restoring balance in dopaminergic nerve terminals. The 5-HT2A receptor mRNA levels were intermediate in the nucleus accumbens [56] and were mostly observed in spiny projecting neurons responsible principally for movement control [57]. Their density was not changed by the acute administration of either psilocybin or ketamine in our study.

Instead, long-term density changes were detected for 5-HT1A and 5-HT2A receptors in the hippocampus. The decreased density of 5-HT1A receptors induced by psilocybin may be beneficial due to the sensitization of these receptors in depression. In contrast, the increased levels of 5-HT2A receptors after the higher dose of psilocybin may reduce the functional deficit of this subtype observed in depression [58] and be related to the increased synaptogenesis found in the hippocampus of the pig brain [59]. Interestingly, the decreased density of 5-HT2A receptors in the hippocampus and prefrontal cortex of the pig brain found by Raval et al. [59] supports our findings after the lower dose of psilocybin. The ketamine-induced increase in 5-HT1A and 5-HT2A receptor density in the hippocampus observed in our work was also evidenced for the other non-competitive antagonists of NMDA receptor [60].

The changes in the levels of neurotransmitters in the limbic system provide the neurobiological background for the psilocybin effect on stress and anxiety, which translates into the antidepressant effect. In our study, psilocybin affected animal behavior in the open field test. The data demonstrated a marked anxiolytic effect of psilocybin in the acute phase and 24 h post-treatment, as shown by increased center penetration and decreased exploration of the peripheral zone of the open field. Instead, ketamine was not effective, or its weak anxiolytic effect was observed only 24 h after treatment. Furthermore, compared to psilocybin, ketamine significantly affected exploration, resulting probably from the stronger (compared to psilocybin) stimulation of DA levels in the nucleus accumbens. The anxiolytic effect of psilocybin observed in the acute phase (though 1 h after administration) may be caused by the reduction in motor activity; however, the increased exploration of the central zone of the open field persisted until 24 h and was no longer accompanied by suppression of rats’ locomotion. Therefore, based on the results of the open field test, it can be concluded that psilocybin and ketamine decreased animals’ anxiety. However, the outcome of the elevated plus maze (EPM) test (Appendix A) contradicts this hypothesis. The lack of drugs’ effect in the EPM test in our hands seems to be caused by the preexposure of animals to an open field test and the drugs’ availability related to the time difference between both tests. A decreased anxiety-like behavior in EPM was evidenced by Hibicke et al. [16] and observed seven days after the administration of single doses of psilocybin and ketamine. However, these authors used Wistar–Kyoto (WKY) rats, a validated model for the study of depression and anxiety, while our experiments were performed in naïve Wistar rats. Thus, the presented data provide insight into symptomatology occurring in healthy human subjects or recreational users of psychedelics. The forced swimming test (FST) was performed, and data were published in our earlier work [23]. While the drugs did not produce an antidepressant effect in FST, it is worth noticing that this assay may not be suitable for testing fast-acting antidepressant drugs [61]. The psilocybin effect observed in the open field test linking behavioral symptoms with a modest elevation of DA and 5-HT in the nucleus accumbens and predominance of GABAergic neurotransmission in other regions of the limbic system has therapeutic implications for treating anxiety and mood disorders.

## 4. Materials and Methods

### 4.1. Animals

Adult male Wistar–Han rats (280–350 g; Charles River, Göttingen, Germany) were used in all experiments. The animals were initially acclimatized and housed (6 per cage) in environmentally controlled rooms (ambient temperature 23 ± 1 °C, humidity 55 ± 10%, and 12:12 light:dark cycle). Rats were handled once daily before the beginning of the experiments; an enriched environment was not applied. The animals had free access to tap water and typical laboratory food (VRF 1, Special Diets Services, Witham, UK). All animal use procedures were conducted in strict accordance with European regulations for animal experimentation (EU Directive 2010/63/EU on the Protection of Animals Used for Scientific Purposes). The 2nd Local Institutional Animal Care and Use Committee (IACUC) in Kraków, Poland, approved the experimental protocols for Experimentation on Animals (permit numbers: 112/2021, app. 8 April 2021; 324/2021, app. 20 October 2020 and 79/2022, app. 10 March 2022).

### 4.2. Drugs and Reagents

Ketamine hydrochloride was purchased from Tocris/Bio-Techne (Warsaw, Poland) and psilocybin was synthesized at the Department of Medicinal Chemistry of the Maj Institute of Pharmacology using the method described by Shirota et al. [62]; both were dissolved in sterile water. All solutions were made fresh on the day of the experiment. The dose of ketamine (10 mg/kg) was based on a report by Popik et al. [63], while doses of psilocybin (2 and 10 mg/kg) were based on work by Jefsen et al. [61]. Psilocybin was given subcutaneously (sc) while ketamine was given intraperitoneally (ip) in the volume of 2 mL/kg. The control group was treated with 0.9% NaCl solution. Ketamine, xylasine hydrochlorides and sodium pentobarbital used for anesthetizing the animals came from Biowet Puławy (Puławy, Poland). All necessary chemicals of the highest purity used for analysis by high-performance liquid chromatography (HPLC) were obtained from Merck (Warsaw, Poland). O-phthalaldehyde (OPA) from Sigma-Aldrich (Poznań, Poland) was used for the derivatization of glutamate and GABA to electroactive compounds. The reagents used in immunohistochemistry were purchased from Sigma-Aldrich (Poznań, Poland), Vector Laboratories (Burlingame, CA, USA), and Proteintech (Manchester, UK).

### 4.3. Brain Microdialysis

Ketamine and xylazine (75 and 10 mg/kg, respectively) were injected intramuscularly to anesthetize the animals. Microdialysis probes (MAB 4.15.3 Cu and MAB 4.15.2 Cu, AgnTho’s AB, Lidingö, Sweden) were implanted into the following brain structures using the determined coordinates (mm): nucleus accumbens AP +1.6, L +1.0, V −8.0, hippocampus AP −5.8, L 4.5, V −5.0 and amygdala AP −3.1, L +4.5, V −8.0 from the dura [64]. Seven days after implantation, probe inlets were connected to a syringe pump (BAS, West Lafayette, IN, USA) which delivered artificial cerebrospinal fluid composed of (mM): 147 NaCl, 4 KCl, 2.2 CaCl_2_·2H_2_O, 1.0 MgCl_2_ at a flow rate of 2 µL/min. Five baseline samples were collected every 20 min after the washout period of 2 h. The respective drugs were administered, and dialysate fractions were collected for the next 240 min. As the experiment ended, the rats were terminated, and their brains underwent histological examination to validate probe placement. The graphical presentation of probes placement is shown in the Appendix A.

### 4.4. Extracellular Concentration of DA, 5-HT, Glutamate, GABA and Acetylcholine

Extracellular DA and 5-HT levels were analyzed using an Ultimate 3000 System (Dionex, Sunnyvale, CA, USA), electrochemical detector Coulochem III (model 5300; ESA, Washington, DC, USA) with a 5020 guard cell, a 5040 amperometric cell, and a Hypersil Gold C18 analytical column (3 μm, 100 × 3 mm; Thermo Fisher Scientific, Waltham, MA, USA). The details of the method have been described elsewhere [65,66]. The chromatographic data were processed by a Chromeleon v.6.80 (Dionex, USA) software package run on a personal computer. The limit of detection of DA and 5-HT in dialysates was 0.02 and 0.04 pg/µL, respectively.

Glutamate and GABA levels in the extracellular fluid were measured by HPLC with electrochemical detection after the derivatization of samples with OPA/sulfite reagent to form isoindole–sulfonate derivatives as previously described [65,66]. The data were processed using Chromax 2005 (Pol-Lab, Warsaw, Poland) software on a personal computer. The limit of detection of glutamate and GABA in dialysates was 0.003 ng/µL and 0.64 pg/µL, respectively.

Extracellular levels of ACh were analyzed by UHPLC with electrochemical detection. The ACh analysis is based on ion-pairing HPLC separation, which is followed by the on-line enzymatic conversion of ACh to hydrogen peroxide and detection on a Pt working electrode (SenCell with 2 mm Pt working electrode) and HyREF reference electrode at the potential of 200 mV. Chromatography was performed using the ALEXYS Neurotransmitter Analyzer, a DECADE Elite electrochemical detector, AS 110 Autosampler, and LC 110 pump (Antec Leyden B. V., Zoeterwoude, The Netherlands). ACh as positively charged was separated on an Acquity UPLC HSS T3 analytical column (1.8 μm, 1 × 50 mm; Waters, Milford, MA, USA). After separation, ACh passed through an immobilized enzyme reactor AChE/ChOx IMER (AC-ENZYM II, 1 × 4 mm, Eicom, Kyoto, Japan). The mobile phase was composed of 50 mM monosodium orthophosphate buffer adjusted to pH 7.8, 0.5 mM Na_2_EDTA, 2.8 g/L 1-octanesulfonic acid sodium salt and 0.5 mM tetramethylammonium chloride. The flow rate during analysis was set to 0.05 mL/min. The chromatographic data were processed by CLARITY v.6.2.0.208 (DataApex Ltd., Prague, Czech Republic) chromatography software run on a personal computer. The detection limit of ACh in dialysates was 0.008 nM.

### 4.5. Western Blotting

The Western blot procedure was performed as described by Wojtas et al. [23]. The hippocampus and nucleus accumbens separated from the brains in anatomical borders were homogenized (TissueLyser, Retsch, Munich, Germany) in lysis buffer (PathScan Sandwich ELISA Lysis Buffer, Cell Signaling, Denver, CO, USA). A Bicinchoninic Acid Kit (Sigma-Aldrich, Poznań, Poland) was used to determine protein concentrations. Protein extracts (20 μg of protein per lane for 5-HT1A, 40 μg protein per lane for 5-HT2A and 40 μg protein per lane for D2 analysis) were separated on a 7.5% SDS-PAGE gel and transferred to nitrocellulose membranes using an electrophoretic transfer system (Bio-Rad Laboratories, Hercules, CA, USA). The membranes were then stained with Ponceau S to confirm gel transfer and cut into three parts: the lower portion was used for GAPDH protein assessment, and the proteins with molecular weights greater than 37 kDa were determined from the next portions of the membrane. The blots were washed, and non-specific binding sites were blocked with 5% albumin (Bovine Serum Albumin; Sigma-Aldrich) and blocking reagent (Lumi Light Western Blotting kit, Roche, Basel, Switzerland) in Tris-buffered saline (TBS) for 1 h at room temperature. Then, the blots were incubated overnight at 4 °C with the following primary antibodies: rabbit anti-5-HT1A (1:1000; ab85615, Abcam, Cambridge, UK), rabbit anti-5-HT2A (1:500; ab216959, Abcam), rabbit anti-D2 (1:500; AB5084P, Sigma-Aldrich) and rabbit anti-GAPDH (1:10,000; 14C10, 2118, Cell Signaling Technology, Danvers, MA, USA). The peroxidase-conjugated secondary anti-rabbit IgG antibody (1:1000, Roche) was used to detect immune complexes (incubation for 1 h at room temperature). Blots were visualized using enhanced chemiluminescence (ECL, Lumi-LightPlus Western Blotting Kit, Roche) and scanned using a luminescent image analyzer (LAS-4000, Fujifilm, Boston, MA, USA). The molecular weights of immunoreactive bands were calculated based on the migration of molecular weight markers (Bio-Rad Laboratories) using Multi Gauge V3.0 (Fujifilm) software. The levels of analyzed proteins were normalized to GAPDH protein. Blots from all experiments are shown in Appendix A.

### 4.6. Open Field Test

The open field test was performed according to the modification of the procedure described by Rogóż and Skuza [67]. A round black arena (1 m in diameter) was virtually divided into eight radiant sections formed by lines intersecting the center of the field. The test was conducted in a dimly lit room, except for the middle of the arena, which was illuminated by a 75 W light bulb placed 75 cm above. Rats were placed in the middle of the arena 60 min after the drug injection. Their behavior was recorded for 5 min. The exploration was quantified with the following parameters: time of walking, number of line crossings reflecting ambulatory distance, episodes of looking under the edge of the field (peeping), number of grooming events, number of rearings as vertical activity and time spent in the central zone.

### 4.7. Statistical Analysis

Drug effects on DA, 5-HT, ACh, GABA and glutamate release in the brain regions were analyzed with repeated measures ANOVA on normalized responses followed by Tukey’s post hoc test. All obtained data were presented as a percentage of the basal level, which was assumed to be 100%. Total effects expressed as area under the curve (AUC) and GABA/GLU ratio were analyzed with one-way ANOVA followed by Tukey’s post hoc test. The results obtained in the open field test and Western blotting data were analyzed with one-way ANOVA followed by Tukey’s post hoc test. The differences were considered significant if *p* < 0.05. The detected outliers were removed from the data set using Grubb’s test. All statistical analyses were conducted using STATISTICA v.10 (StatSoft Inc. 1984–2011 (Tulsa, OK, USA)) and GraphPad Prism v.5.00 (GraphPad Software Inc., La Jolla, CA, USA).

## 5. Conclusions

In conclusion, the presented data have mechanistic significance and show the implications of the psilocybin impact on neurotransmitter levels for the therapy of depression and anxiolytic disorders. The increased dopaminergic and serotonergic neurotransmission in the nucleus accumbens and predominant GABAergic signaling in limbic brain regions link the current findings with brain abnormalities observed in clinical studies.

In our behavioral and neurochemical experiments, ketamine has served as a comparative substance to psilocybin with a confirmed fast-acting antidepressant effect. Unfortunately, ketamine together with xylazine was also used to anaesthetize animals for the implantation of probes in microdialysis experiments. This treatment may potentially impact psilocybin and ketamine action in their response on neurotransmitter release in limbic brain regions. However, animals were left for seven days after guide cannulas implantation to recover. Thus, the actual microdialysis experiments were performed in freely moving animals after a time period sufficient for alleviating the risk of ketamine affecting results. Furthermore, comparisons with alternative anesthesia have been made, and there was no difference in response to psilocybin and ketamine given in the lower dose. It should be noted however, that incidents of mortality were recorded for some animals.

Our study was performed in male rats, but it is known that females are more sensitive to the psychoplastogenic effects of psilocybin [68], while a greater impairment in PPI and locomotor activity was observed in male rats [69]. The problem of sex differences in response to psychedelic drugs requires further research.

## Figures and Tables

**Figure 1 ijms-25-00100-f001:**
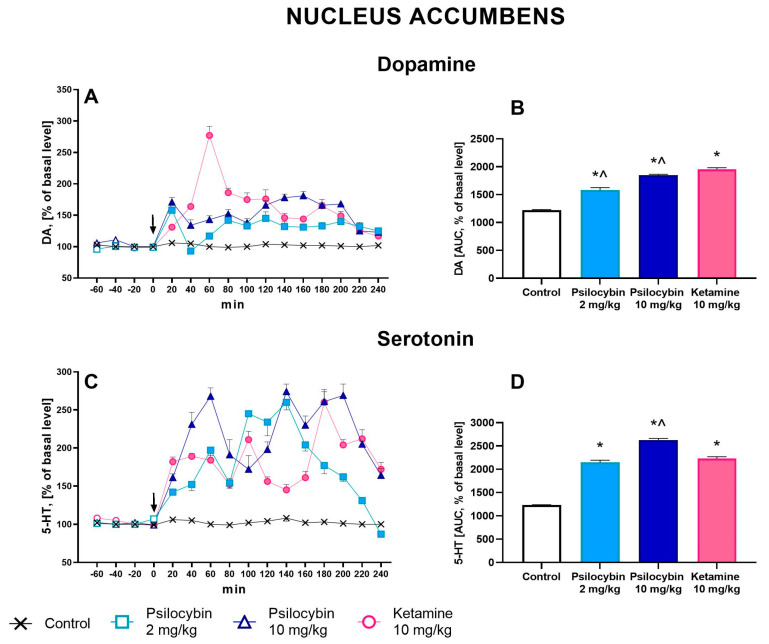
The time-course (**A**,**C**) and total (**B**,**D**) effect of psilocybin (2 and 10 mg/kg) and ketamine (10 mg/kg) on the dopamine (DA) and serotonin (5-HT) levels in the rat nucleus accumbens. The total effect was calculated as an area under the concentration–time curve (AUC) and is expressed as a percentage of the basal level. Values are the mean ± SEM (*n* = 6–8) for each neurotransmitter. The drug injection is indicated with an arrow. The baseline values of DA and 5-HT are presented in the Appendix A. Filled symbols or * show statistically significant differences (*p* < 0.001) between control and drug treatment groups, ^ (*p* < 0.05) show significant differences between ketamine and psilocybin and ketamine groups as estimated by repeated measures ANOVA (time-course) or one-way ANOVA (total effect) followed by Tukey’s post hoc test.

**Figure 2 ijms-25-00100-f002:**
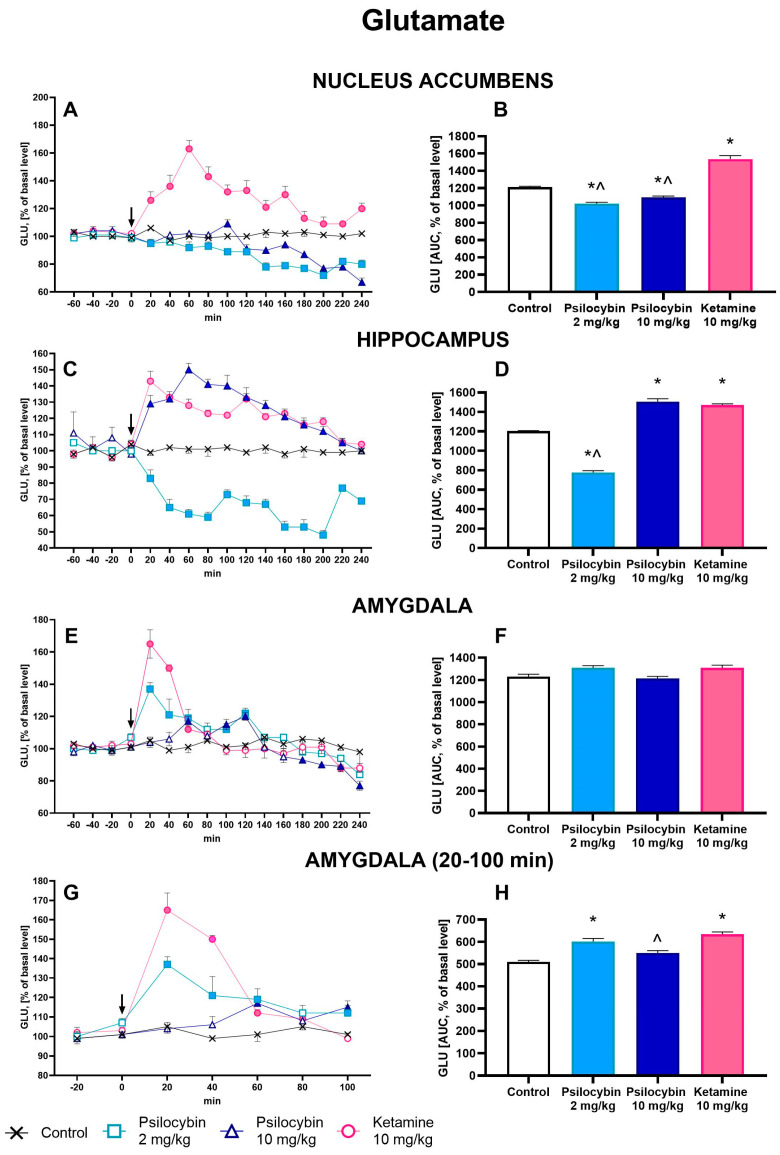
The time course (**A**,**C**,**E**,**G**) and total (**B**,**D**,**F**,**H**) effect of psilocybin (2 and 10 mg/kg) and ketamine (10 mg/kg) on the extracellular levels of glutamate (GLU) in the rat nucleus accumbens, hippocampus and amygdala. The total effect was calculated as an area under the concentration–time curve (AUC) and is expressed as a percentage of the basal level. Values are the mean ± SEM (*n* = 6–9 in nucleus accumbens groups, *n* = 6–7 in hippocampus groups, *n* = 5–6 in amygdala groups. The drug injection is indicated with an arrow. The baseline values of glutamate in dialysates from all brain regions are given in the Appendix A. Filled symbols or * (*p* < 0.001) show statistically significant differences between control and drug treatment groups, ^ (*p* < 0.001) show statistical differences between psilocybin and ketamine groups as estimated by repeated measures ANOVA (time-course) or one-way ANOVA (total effect) followed by Tukey’s post hoc test.

**Figure 3 ijms-25-00100-f003:**
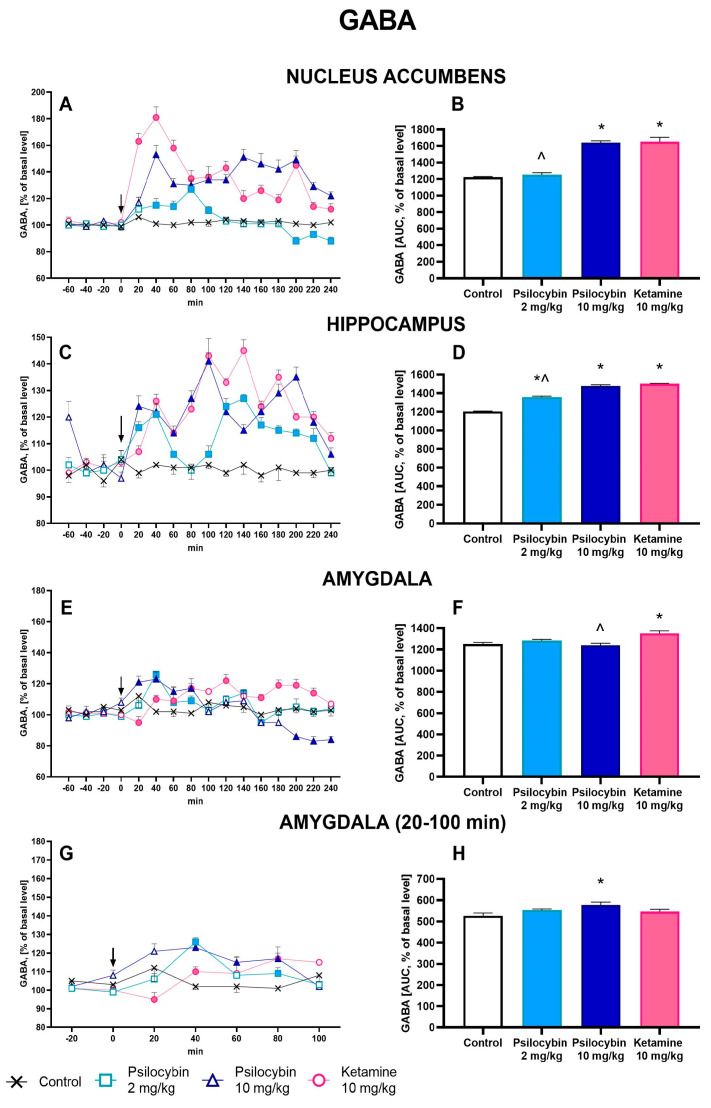
The time-course (**A**,**C**,**E**,**G**) and total (**B**,**D**,**F**,**H**) effect of psilocybin (2 and 10 mg/kg) and ketamine (10 mg/kg) on the extracellular levels of γ-aminobutyric acid (GABA) in the rat nucleus accumbens, hippocampus and amygdala. The total effect was calculated as an area under the concentration–time curve (AUC) and is expressed as a percentage of the basal level. Values are the mean ± SEM (*n* = 6–9 in nucleus accumbens groups, *n* = 6–7 in hippocampus groups, *n* = 5–6 in amygdala groups). The drug injection is indicated with an arrow. The baseline values of GABA in dialysates from all brain regions are given in the Appendix A. Filled symbols or * (*p* < 0.001) show statistically significant differences between control and drug treatment groups, ^ (*p* < 0.001) shows statistical differences between psilocybin and ketamine groups as estimated by repeated measures ANOVA (time-course) or one-way ANOVA (total effect) followed by Tukey’s post hoc test.

**Figure 4 ijms-25-00100-f004:**
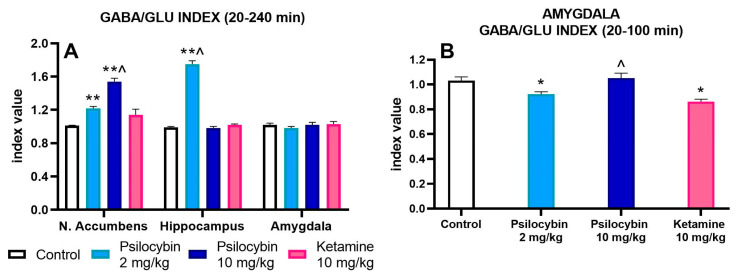
The AUC GABA/GLU ratio during 20240 min of collection period after administration of psilocybin (2 and 10 mg/kg) and ketamine (10 mg/kg) in the rat nucleus accumbens, hippocampus and amygdala (**A**) and during 20–100 min of collection period in the amygdala (**B**). Values are the mean ± SEM (*n* = 6–9 in the nucleus accumbens groups, *n* = 6–7 in hippocampus groups and *n* = 5–6 in amygdala groups) as estimated by one-way ANOVA followed by Tukey’s post hoc test. * *p* < 0.02, ** *p* < 0.001 indicate statistically significant differences between control and drug treatment groups, ^ (*p* < 0.001) show statistical differences between psilocybin and ketamine groups.

**Figure 5 ijms-25-00100-f005:**
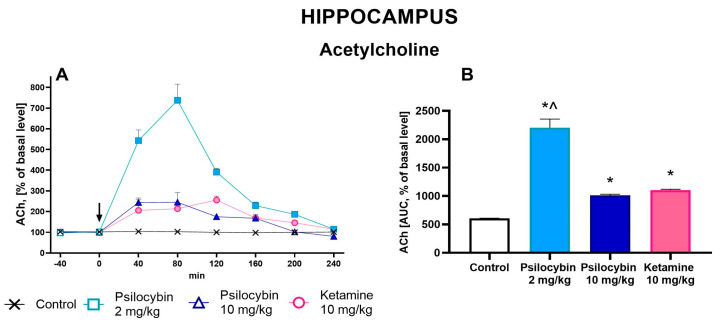
The time-course (**A**) and total (**B**) effect of psilocybin (2 and 10 mg/kg) and ketamine (10 mg/kg) on the extracellular acetylcholine (ACh) levels in the rat hippocampus. The total effect was calculated as an area under the concentration–time curve (AUC) and is expressed as a percentage of the basal level. Values are the mean ± SEM (*n* = 5–7). The drug injection is indicated with an arrow. The baseline values of ACh in dialysates from the hippocampus are given in Appendix A. Filled symbols or * show statistically significant differences (*p* < 0.001) between control and drug treatment groups, ^ (*p* < 0.001) show statistical difference between psilocybin lower dose and ketamine groups as estimated by repeated measures ANOVA (time-course) or one-way ANOVA (total effect) followed by Tukey’s post hoc test.

**Figure 6 ijms-25-00100-f006:**
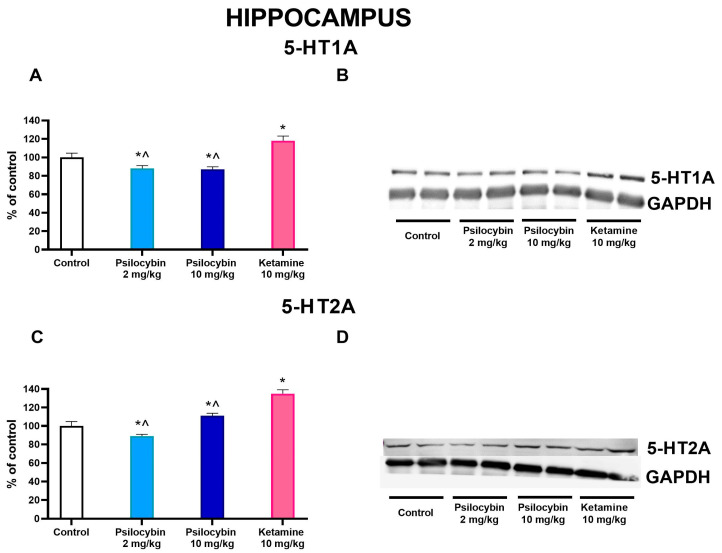
The levels of 5-HT1A receptor (**A**) and 5-HT2A receptor (**C**) in the rat hippocampus were estimated 7 days after psilocybin (2 and 10 mg/kg) or ketamine (10 mg/kg) administration. The data are shown as percentages of the levels of the appropriate control groups. Each data point represents the mean ± SEM (*n* = 10). * *p* < 0.001 vs. control group, ^ (*p* < 0.001) shows statistical difference between psilocybin and ketamine groups (one-way ANOVA followed by Tukey’s post hoc test). Representative examples of photomicrographs of the immunoblots using 5-anti-HT1A and anti-5-HT2A antibodies ((**B**,**D**), respectively).

**Figure 7 ijms-25-00100-f007:**
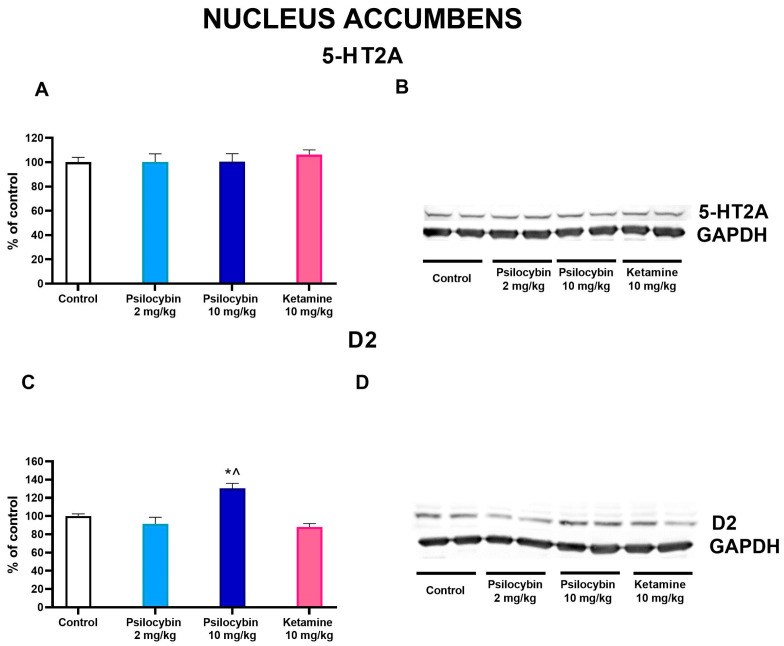
The levels of 5-HT2A receptor (**A**) and D2 receptor (**C**) in the rat nucleus accumbens were estimated 7 days after psilocybin (2 and 10 mg/kg) or ketamine (10 mg/kg) administration. The data are shown as percentages of the levels of the appropriate control groups. Each data point represents the mean ± SEM (*n* = 10). * *p* < 0.0001 vs. control group, ^ (*p* < 0.001) show statistical difference between psilocybin and ketamine groups (one-way ANOVA followed by Tukey’s post hoc test). Representative examples of photomicrographs of the immunoblots using 5-anti-HT2A and anti-D2 antibodies ((**B**,**D**), respectively).

**Figure 8 ijms-25-00100-f008:**
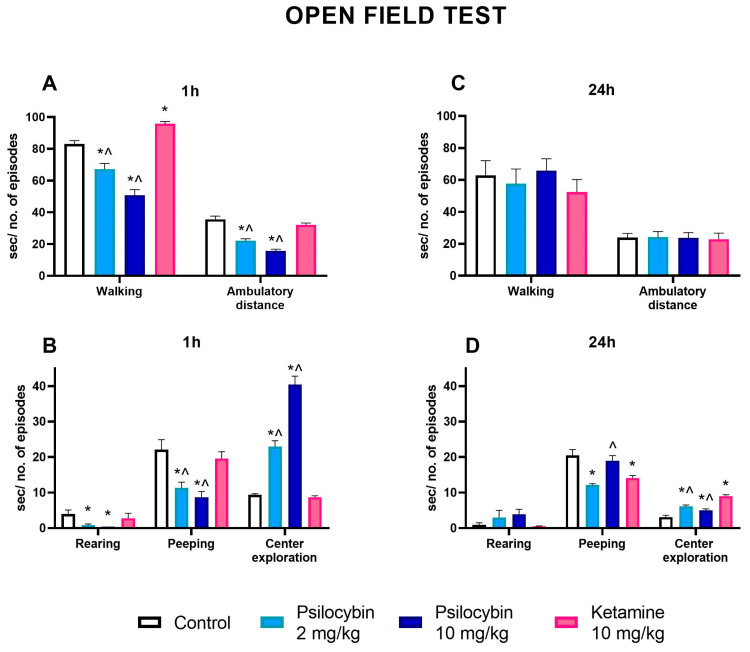
The effect of psilocybin (2 and 10 mg/kg) and ketamine (10 mg/kg) on rat behavior in the open field test. The time spent on walking and ambulatory distance at 1 h (**A**) and 24 h (**C**) after administration, the number of rearing and peeping episodes and time spent in the center at 1 h (**B**) and 24 h (**D**), respectively. Values are the mean ± SEM (*n* = 10). * *p* < 0.01 compared to the control, ^ (*p* < 0.05) show statistical difference between psilocybin and ketamine groups (one-way ANOVA followed by Tukey’s post hoc test).

## Data Availability

All data are contained within the article and Appendix A.

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
