# Peer review of "Limbic System Response to Psilocybin and Ketamine Administration in Rats: A Neurochemical and Behavioral Study"

_ijms, 2023, doi:10.3390/ijms25010100_

Round 1
Reviewer 1 Report
Comments and Suggestions for Authors
The study investigates the impact of varying doses of psilocybin and ketamine on dopamine, serotonin, glutamate, GABA, and acetylcholine levels in the rat's nucleus accumbens, hippocampus, and amygdala. Results indicate time and concentration-dependent alterations. Psilocybin and ketamine also influence serotonin receptor expression in the hippocampus, and the high psilocybin dose affects dopamine receptors in the nucleus accumbens. These findings shed light on the effects of emerging antidepressants like psilocybin and their comparative advantages over potentially addictive substances such as ketamine, contributing to our understanding of novel treatment possibilities.
Minor revisions
- The authors should be more anatomically precise in which area of the nucleus accumbens and amygdala they selected for the microdialysis and western blots.
- Considering that ketamine is used for anesthesia across all rat groups, the authors should discuss the potential impact of this choice on subsequent results, especially given that ketamine serves as the comparative drug to psilocybin.
- The figures exhibit poor quality upon zooming in, necessitating improvement.
- While conducting post hoc tests on the AUC graphs, the authors do not indicate differences between psilocybin and ketamine groups, aside from distinctions with the control group. The panel should explicitly indicate any statistically significant differences between psilocybin and ketamine groups.
- The authors should discuss why the 10mg/kg dose of psilocybin fails to elevate glutamate levels in the amygdala, despite the observed increase with a 2mg/kg dose.
- Given that open field and elevated plus maze tests commonly assess anxiety-like behaviors, the authors should clarify their rationale for not including a depression-like test such as the forced swim test.
Major revisions
- Although the authors conducted the elevated plus maze test, it is not reported in the results nor the discussion. It is important to compare the outcome of this test with the open field and previous literature.
- The study is performed only in male rats. The authors should discuss their choice of not including females and the possible sex effects they might find if added.
- The manuscript lacks clarity regarding the utilization of the same groups of rats for the open field test conducted 2 hours and 24 hours after drug administration. The authors should explicitly specify whether it is the same cohort of rats and, if so, address the potential influence of prior exposure to the open field on subsequent anxiety levels, considering the likelihood of habituation.
- While dopamine receptors are expressed in both the hippocampus and central amygdala, the study does not assess protein levels in these areas. The authors should provide a justification for not adding these areas to their analysis.
Author Response
The study investigates the impact of varying doses of psilocybin and ketamine on dopamine, serotonin, glutamate, GABA, and acetylcholine levels in the rat's nucleus accumbens, hippocampus, and amygdala. Results indicate time and concentration-dependent alterations. Psilocybin and ketamine also influence serotonin receptor expression in the hippocampus, and the high psilocybin dose affects dopamine receptors in the nucleus accumbens. These findings shed light on the effects of emerging antidepressants like psilocybin and their comparative advantages over potentially addictive substances such as ketamine, contributing to our understanding of novel treatment possibilities.
Minor revisions
- The authors should be more anatomically precise in which area of the nucleus accumbens and amygdala they selected for the microdialysis and western blots.
For western blots, the regions in their anatomical borders were isolated. The placement of microdialysis probes is presented in the supplementary material 4.
- Considering that ketamine is used for anesthesia across all rat groups, the authors should discuss the potential impact of this choice on subsequent results, especially given that ketamine serves as the comparative drug to psilocybin.
We were aware of this problem when deciding to use ketamine/xylazine anesthesia. However, animals were left for recovery for seven days after guides implantation, and the actual microdialysis experiment was conducted in freely moving rats. In addition, we have made comparisons with the use of other kinds of anaesthesia (chloral hydrate), and there was no difference in response to psilocybin and ketamine given in the low dose. However, incidents of mortality happened in some animals.
- The figures exhibit poor quality upon zooming in, necessitating improvement.
The figures have been improved.
- While conducting post hoc tests on the AUC graphs, the authors do not indicate differences between psilocybin and ketamine groups, aside from distinctions with the control group. The panel should explicitly indicate any statistically significant differences between psilocybin and ketamine groups.
Significant differences between psilocybin and ketamine groups were indicated in all figures.
- The authors should discuss why the 10mg/kg dose of psilocybin fails to elevate glutamate levels in the amygdala, despite the observed increase with a 2mg/kg dose.
5-HT2A receptors are highly expressed on pyramidal cells and both parvalbumin and somatostatin GABAergic interneurons in the amygdala, which suggests that the effect of their activation by psilocybin higher dose may be suppressive (Bombardi C., Di Giovanni G. (2013) doi: 10.1007/s00221-013-3512-6. In fact, psilocybin’s higher dose increased GABA release in the amygdala.
- Given that open field and elevated plus maze tests commonly assess anxiety-like behaviors, the authors should clarify their rationale for not including a depression-like test such as the forced swim test.
The forced swimming test was performed, and this data was already published in our earlier work (Wojtas et al. Int. J. Mol. Sci. 2022, 23, 6713, doi.org/10.3390/ijms23126713). While the drugs did not produce an antidepressant effect in the FST, it is worth noticing that this assay may not be suitable for testing fast-acting antidepressant drugs (Jefsen et al., 2019).
Major revisions
- Although the authors conducted the elevated plus maze test, it is reported in neither the results nor the discussion. It is important to compare the outcome of this test with the open field and previous literature.
The results conducted in the EPM test were mentioned in the Results section, and respective notification was added to the Discussion.
- The study is performed only in male rats. The authors should discuss their choice of not including females and the possible sex effects they might find if added.
We are aware of sex differences observed in clinical and preclinical studies, such as greater sensitivity of females to the psychoplastogenic effect of psilocybin (Shao et al. 2021) and greater impairment in PPI in male rats (Tyls et al. 2016), but these are our future directions.
- The manuscript lacks clarity regarding the utilization of the same groups of rats for the open field test conducted 2 hours and 24 hours after drug administration. The authors should explicitly specify whether it is the same cohort of rats and, if so, address the potential influence of prior exposure to the open field on subsequent anxiety levels, considering the likelihood of habituation.
The open field test was conducted 1 h and 24 h after drug administration in the same cohort of rats. The difference in response at 1 h and 24 h may result from the drug wash out or habituation of animals. This was notified in the Discussion.
- While dopamine receptors are expressed in both the hippocampus and central amygdala, the study does not assess protein levels in these areas. The authors should provide a justification for not adding these areas to their analysis.
The levels of dopamine receptors were conducted in the nucleus accumbens as this area is involved in the reward system. We observed increased DA release in the nucleus accumbens by psilocybin and ketamine in the presence of the drugs. D2 receptors play a regulatory role in the release of DA from neuronal terminals and changes in their density may affect the reward system. The increase in their density by psilocybin seems to be evidence of regulatory mechanisms activation to maintain the normal function of the reward system.
Unfortunately, we were not able to measure DA release in the hippocampus and amygdala since the basal levels of DA and 5-HT were below the detection limit of our HPLC systems. DA receptors are expressed also in hippocampus and amygdala, but also at the very low levels.
Reviewer 2 Report
Comments and Suggestions for Authors
The article titled “Limbic system response to psilocybin and ketamine administration in rats: A neurochemical and behavioral study” it is well structured and written. The methodology used is adjusted to the objectives described and the discussion is adequately focused on the results obtained. I only suggest one minor modification.
In relation to the results, I believe that the neurotransmitter levels have only been represented when it has been possible to detect a basal level in a certain area. If this is so, I think it would clarify a lot to express it in the section that the authors considered.
Author Response
The article titled “Limbic system response to psilocybin and ketamine administration in rats: A neurochemical and behavioral study” it is well structured and written. The methodology used is adjusted to the objectives described and the discussion is adequately focused on the results obtained. I only suggest one minor modification.
In relation to the results, I believe that the neurotransmitter levels have only been represented when it has been possible to detect a basal level in a certain area. If this is so, I think it would clarify a lot to express it in the section that the authors considered.
Exactly, this is a reason not analyzing of monoaminergic neurotransmitters in hippocampus and amygdala since the basal levels of DA and 5-HT were below detection limit of our HPLC systems.
Reviewer 3 Report
Comments and Suggestions for Authors
Wojtas et al utilized microdialysis and western blotting from multiple brain regions in rats to assess alterations in neurotransmission after psilocybin and ketamine treatment. Lack of effective antidepressant treatments causes tremendous economic burden as well as suffering for depression patients and their families. Ketamine, the FDA-approved rapid acting antidepressant, provided therapeutic benefits and novel directions for future novel treatments. Psilocybin is more recently being studied as candidate for next-generation antidepressant treatment, even less is known for its mechanism of inducing therapeutic effects. This manuscript provides evidence for potential alterations within the limbic areas that may correlate with beneficial effect induced by psilocybin. However, there are several major and minor concerns remained to be resolved, listed as follows.
1. All data provided in the manuscript is descriptive, which makes it difficult to make the argument that observed results are important for the antidepressant effects induced by psilocybin.
2. There are discrepancies between the timeline of behavioral phenotypes and neuro-biochemical measurements. Without results obtained at the consistent timepoints, it is presumptuous to assume correlation.
3. Quality of western blots needs to be significantly improved, given the internal control bands cannot be shown consistently on different blots.
4. In Fig.7, which is the only figure regarding behavioral effects after ketamine and psilocybin treatments, the supposedly antidepressant effect at 24h with ‘increased’ center exploration is more likely to be caused by drastic reduced exploration in the control group. Therefore, this ‘antidepressant’ effect induced by psilocybin and ketamine cannot be concluded with the data provided.
5. Rats used for microdialysis experiments were anesthetized with ketamine/xylazine, given that ketamine was also used for treatment and is known to alter neurotransmission, results obtained from those experiments are confounded by ketamine’s effect. If alternative anesthetics are not available, the authors need to address this caveat in the manuscript.
Comments on the Quality of English LanguageEnglish needs to be edited for clearly understanding
Author Response
Wojtas et al. utilized microdialysis and western blotting from multiple brain regions in rats to assess alterations in neurotransmission after psilocybin and ketamine treatment. Lack of effective antidepressant treatments causes tremendous economic burden as well as suffering for depression patients and their families. Ketamine, the FDA-approved rapid-acting antidepressant, provided therapeutic benefits and novel directions for future novel treatments. Psilocybin is more recently being studied as a candidate for next-generation antidepressant treatment; even less is known about its mechanism of inducing therapeutic effects. This manuscript provides evidence for potential alterations within the limbic areas that may correlate with the beneficial effect induced by psilocybin. However, several major and minor concerns that remained to be resolved are listed as follows.
- All data provided in the manuscript is descriptive, which makes it difficult to make the argument that observed results are important for the antidepressant effects induced by psilocybin.
I do not agree that the data provided are descriptive. At this stage of knowledge on the mechanism of antidepressant effect of psychedelics, it is proven in in vitro and in vivo models that psychedelics induce plastic changes enabling TrkB/mTOR dependent signaling and that 5-HT2A receptor located at deep pyramidal cells in layer V of the prefrontal cortex is crucial for this effect as well as for subjective experience. Downstream effects are glutamatergic projections executing control of subcortical regions, which participate in various cognitive and emotional functions. It is not clear how these projections impact subcortical circuits. Our findings provide neurobiological insight into synaptic events such as neurotransmitters release and receptor level. Importantly, psilocybin interaction with 5-HT1A receptors, which colocalize with 5-HT2A type, seem to regulate in a distinct way cells excitability depending on receptors location and their density across brain areas. These complex interactions are essential for understanding how psychedelics impact behavior and what is their safety limit. In light of latest research showing robust c-Fos expression across the wide areas of the brain or other immediate early genes, i.e. EGR1 by single doses of psilocybin (Davoudian et al. 2022; Liu et al. 2023), our data integrate these findings with neurotransmitter function and behavioural effects.
- There are discrepancies between the timeline of behavioral phenotypes and neuro-biochemical measurements. Without results obtained at the consistent timepoints, it is presumptuous to assume correlation.
All our experiments in which the release of neurotransmitters was monitored show acute drug effects, thus in the presence of the drugs. Behavioral effect in the open field test is also presented 1 hr after drug injections and is correlated with all microdialysis data. As structural neural plasticity is observed within 24 h and longer, we were interested in whether there are any behavioral phenotypes and whether some adaptatory changes of receptors in the drug absence may be observed in the long term as a result of drug impact on neurotransmitters release. Therefore, 5-HT2A and 5-HT1A receptors’ density was investigated in the hippocampal and D2 and 5-HT2A in the accumbal tissue.
- Quality of western blots needs to be significantly improved, given the internal control bands cannot be shown consistently on different blots.
In the supplement materials and Figs 6 and 7, examples of original images of western blots are presented. Each blot was taken under several exposure conditions, and images providing the best optimal saturation of internal or investigated protein bands were analyzed. Each blot was evaluated separately, and each internal control band (GAPDH) was used to normalise a suitable band of investigated proteins. Thus, in our opinion, there is no need to have consistent internal control bands on different blots, since each blot was probed independently.
- In Fig.7, which is the only figure regarding behavioral effects after ketamine and psilocybin treatments, the supposedly antidepressant effect at 24h with ‘increased’ center exploration is more likely to be caused by drastic reduced exploration in the control group. Therefore, this ‘antidepressant’ effect induced by psilocybin and ketamine cannot be concluded with the data provided.
I agree with the reviewer, the longer exploration of animals in the center of the field may be caused by the reduction in motor activity of animals treated with psilocybin observed 1h after administration (but not in the control group!). I have added an explanation when referring to the data of the open field. However, the increased exploration of the central zone of the field persisted for 24 h, but there was no motor activity decrease of rats observed. This result allows us to preserve the conclusion that psilocybin and ketamine decreased animal anxiety.
- Rats used for microdialysis experiments were anesthetized with ketamine/xylazine, given that ketamine was also used for treatment and is known to alter neurotransmission, results obtained from those experiments are confounded by ketamine’s effect. If alternative anesthetics are not available, the authors need to address this caveat in the manuscript.
We were aware of this problem when deciding to use ketamine/xylazine anesthesia. However, animals were left for recovery for seven days after guides implantation and the actual microdialysis experiment was conducted in freely moving rats. In addition, we have made comparisons with the use of other kinds of anaesthesia (chloral hydrate), and there was no difference in response to psilocybin and ketamine given in the low dose. However, incidents of mortality happened in some animals.
Comments on the Quality of English Language
English needs to be edited for clearly understanding
English editing was corrected in this version of the manuscript.
Round 2
Reviewer 1 Report
Comments and Suggestions for Authors
The authors should add a limitations section in the discussion with the answers to these two comments:
- Considering that ketamine is used for anesthesia across all rat groups, the authors should discuss the potential impact of this choice on subsequent results, especially given that ketamine serves as the comparative drug to psilocybin.
- The study is performed only in male rats. The authors should discuss their choice of not including females and the possible sex effects they might find if added.
Author Response
The authors should add a limitations section in the discussion with the answers to these two comments:
- Considering that ketamine is used for anesthesia across all rat groups, the authors should discuss the potential impact of this choice on subsequent results, especially given that ketamine serves as the comparative drug to psilocybin.
- The study is performed only in male rats. The authors should discuss their choice of not including females and the possible sex effects they might find if added.
As Reviewer suggests, answers to above comments were added as study limitations at the end of discussion.
Reviewer 3 Report
Comments and Suggestions for Authors
I would suggest the authors to include their response to my previous question about using ketamine/xylazine as anesthetics and then using ketamine as an antidepressant treatment in their behavior experiments into the manuscript, especially that they had tried with an alternative anesthetic agent.
Regarding the western blot showing in now Fig 6 B and D, showing the wavy GAPDH bands as the representative blots indicates the experiment itself may not be performed under the optimal conditions. Though I appreciate the authors' effort in showing the original blots, this quality is not satisfying for publication.
Author Response
I would suggest the authors to include their response to my previous question about using ketamine/xylazine as anesthetics and then using ketamine as an antidepressant treatment in their behavior experiments into the manuscript, especially that they had tried with an alternative anesthetic agent.
As suggested, the response to this question is included in the final section of the manuscript (as study limitations).
Regarding the western blot showing in now Fig 6 B and D, showing the wavy GAPDH bands as the representative blots indicates the experiment itself may not be performed under the optimal conditions. Though I appreciate the authors' effort in showing the original blots, this quality is not satisfying for publication.
All original blots were shown in supplementary material 1.